# A Fault Diagnosis Model for Tennessee Eastman Processes Based on Feature Selection and Probabilistic Neural Network

Haoxiang Xu , Tongyao Ren, Zhuangda Mo and Xiaohui Yang *

College of Information Engineering, Nanchang University, Nanchang 330031, China
* Correspondence: yangxiaohui@ncu.edu.cn; Tel.: +86-139-7094-1450

**Abstract:** Since the classification methods mentioned in previous studies are currently unable to meet the accuracy requirements for fault diagnosis in large-scale chemical industries, these methods are gradually being eliminated and rarely used. This research offers a probabilistic neural network (PNN) based on feature selection and a bio-heuristic optimizer as a fault diagnostic approach for chemical industries using artificial intelligence. The sample characteristics are initially simplified using heuristic feature selection and support vector machine recursive feature elimination (SVM-RFE). Using PNN as the principal classifier of the fault diagnostic model and employing a modified salp swarm algorithm (MSSA) linked with the bio-heuristic optimizer to optimize the hidden smoothing factor ($\sigma$) of PNN further improves the classification performance of PNN. The MSSA introduces the Lévy flight method, greatly enhancing exploration capabilities and convergence speed compared to the standard SSA. To validate the engineering application of the suggested method, a PSO-SVM-REF-MSSA-PNN model is created, and TE process data are utilized in tests. The model's performance is evaluated by comparing its accuracy and F1-score to other regularly used classification models. The results indicate that the data samples selected by PSO-SVM-RFE features simplify and eliminate redundant features more effectively than other feature selection techniques. The MSSA algorithm's optimization capabilities surpass those of conventional optimization techniques. The PNN network is more suitable for fault detection and classification in the chemical industry. The three considerations listed above make it evident that the proposed approach might greatly help identify TE process problems.

**Keywords:** fault diagnosis; probabilistic neural network; TE process; modified salp swarm algorithm; feature selection; SVM-RFE; PSO-SVM

## 1. Introduction

Due to the rapid growth of mechanical automation and artificial intelligence, chemical control systems no longer rely solely on manual operations to manage complicated chemical conditions. However, the products of the chemical process are frequently poisonous, combustible, and explosive due to its tedious procedures and changing environment. Once a breakdown has occurred, the risk is significantly greater than in other businesses, and it is simple to cause massive deaths, environmental degradation, and economic losses [1,2]. In the past few years, academic and business circles around the world have become more interested in chemical process defect detection and diagnosis, which is one of the most important parts of modern chemical systems [3].

Failure in a chemical process is characterized by the deviation of one or more process variables from their normal state. The fault detection and diagnosis technology monitors the entire system operation process and determines the fault type based on state variables with different deviations [4,5]. Currently, defect diagnosis methodologies can be loosely categorized as knowledge-driven, model-driven, and data-driven [6]. However, the system's size continues to increase, and the tightness of the correlation between feature variables increases. Knowledge-driven and model-driven diagnostic procedures can no longer meet

modern industrial systems' speed and accuracy requirements for massive data processing. The data-driven method is based on process data, makes a decision-making model, mimics how the factory actually works, can find and diagnose problems well, and is becoming the standard method.

In the realm of industrial process fault diagnosis, multivariate statistical approaches, such as independent component analysis (ICA), principal component analysis (PCA), and fisher discriminant analysis (FDA), have been frequently utilized [7–10]. As the number of data dimensions increases, however, the complexity of these statistics-based procedures increases exponentially, resulting in a dimensional catastrophe. The gradual application of shallow learning techniques such as support vector machine (SVM), k-nearest neighbor (KNN), and artificial neural network (ANN) converts defect detection and diagnosis problems into classification problems [11–13]. However, they rely on extensive training and fault samples, which are difficult for chemical diagnostic models.

The problem of troubleshooting chemical processes has been of interest to research scholars. Ragab et al. [14] discovered hidden knowledge in industrial datasets by revealing explainable patterns associated with underlying physical phenomena through logical analysis of data (LAD). These patterns are then combined to build a decision model for diagnosing faults during process operations and explaining the potential causes of these faults. Zhang et al. [15] used bidirectional recurrent neural network (BiRNN) to construct fault detection and diagnosis (FDD) models with complex RNN units and demonstrated the effectiveness of implementing BiRNN in chemical process fault diagnosis. Wang et al. [16] proposed an extended deep belief network (EDBN) to use the valuable information in the raw data fully. The raw data are also combined with hidden features as the input to each extended restricted Boltzmann machine (ERBM). Wang et al. [17] used long short-term memory (LSTM) and convolutional neural network (CNN) to extract features separately and then fused the extracted features. The features are further compressed and extracted by using them as the input of a multilayer perceptron so that the final extracted features of the network have both spatial and temporal features, thus improving the diagnostic performance of the network.

In addition, artificial neural networks have promising applications in fault diagnosis, and many different types of artificial neural networks are available for classification tasks. As a supervised network classifier based on the Bayesian minimal risk criterion, the probabilistic neural network (PNN) does not require weight adaption, the learning process is straightforward, the training speed is quick, and it possesses more robustness and fault tolerance. In addition, even with fewer training data, the insensitivity to noisy data maintains an excellent diagnostic accuracy. It has been applied successfully to photovoltaic array fault diagnosis [18], circuit breakers fault diagnosis [19], and distributed generation fault diagnosis [20]. However, PNN has significant limitations, such as a low recognition rate and misclassification due to the usage of the same smoothing factor in the iterative process and a complex network structure when the sample set is large. In recent years, meta-heuristic algorithms, such as grey wolf optimizer (GWO) [21], particle swarm optimization (PSO) [22], and sparrow algorithm (SA) [23], have grabbed the attention of researchers and been widely applied to improve the diagnostic performance of PNN further. However, the recently created salp swarm algorithm (SSA) in 2017 has some advantages over conventional optimization algorithms, including a straightforward theory and a rapid search rate. In addition, it has the unique benefit that exploration and production are balanced by a single parameter ($c_1$) [24,25].

Although research into chemical process fault identification has made tremendous strides thanks to machine learning techniques, there are still some issues to be resolved. Numerous studies have demonstrated that, due to noisy features, computations with all feature sets in chemical processes may not always yield ideal results. Feature selection algorithms can be used to remove these superfluous features. The features election problem in classification can be characterized as "identifying the smallest subset of features from the whole collection of features that achieves the highest classification accuracy." However,

this frequently needs exponential calculation time, which is challenging. To improve classification, researchers employ evolutionary and heuristic methods to feature selection, such as genetic algorithm (GA) [26], ant colony optimization (ACO) [27], particle swarm optimization (PSO) [28], etc. In this feature selection method, the particle swarm algorithm is combined with the support vector machine, which is used to evaluate the fitness value of the particle swarm. This allows for more effective implementation of the feature selection process, as well as improved processing speed and accuracy. When used to classify issues, the method can improve accuracy by 2 to 4% [29].

SVM is not only a data-driven classification technique, but also an excellent machine learning technique. Numerous novel techniques combine data dimensionality reduction with SVM for process monitoring, defect information extraction, and variable elimination. SVM can give their dimension reduction techniques. For example, recursive feature elimination approaches choose the most important features by using accurate category rankings, getting important and useful information from samples, rebuilding samples for classification, and using this method to successfully diagnose chemical faults [30].

This work provides a novel chemical process defect diagnosis model based on Tennessee Eastman (TE) data and prior research expertise. The establishment of the model involves three sequential steps: establishing a two-stage feature selection approach with PSO-SVM and SVM-RFE, updating SSA with Lévy flight, and developing a fault detection method based on PNN with an optimum smoothing factor. The steps are as follows:

1.  When there are nonlinear, high-dimensional TE process datasets, we use a two-stage feature selection method to eliminate duplicate features and reduce memory needs. This makes fault diagnosis more accurate and effective.
2.  The Lévy flight method is included in SSA, and a new algorithm, MSSA, is developed to alleviate SSA's deficiencies, such as its slow convergence speed and propensity to slip into local optimum. The approach can iteratively randomize the leader's position and enhance the optimal global searchability. In addition, it can provide selective updates to followers, which will accelerate convergence.
3.  Using MSSA to optimize the smoothing factor of PNN can improve the reliability, self-correction capability, and accuracy of PNN when dealing with data categorization problems.

The rest of the paper is structured as follows. Section 2 introduces feature selection, machine learning algorithms, and models based on them. Section 3 compares the model proposed in this paper with previous models from three perspectives and demonstrates the superiority of the model proposed in this paper. Section 4 summarizes our contributions and presents our future work.

## 2. Materials and Methods

The efficient combination of feature selection and neural networks is one of the effective ways to deal with high-dimensional and massive data. After long-term theoretical development and practical exploration, it has some unique advantages. In this section, we first review some algorithms and techniques that are prerequisites for our work. Then, based on these algorithms and techniques, we constructed a fault diagnosis model for Tennessee Eastman chemical processes.

### 2.1. Feature Selection Phase

Selecting the most relevant features for the training phase is an essential step in many pattern recognition problems. Therefore, the critical question is how to find the excellent subset of elements matching the data categories to enhance the performance of pattern recognition models. To tackle this challenging task, many feature selection algorithms have been developed. This section first introduces support vector machine recursive feature elimination (SVM-RFE) and feature selection using particle swarm optimization (PSO)-SVM. Then, a two-level feature selection preprocessing model is constructed based on both.

### 2.1.1. Support Vector Machine Recursive Feature Elimination

Guyon et al. [31,32] first suggested the support vector machine recursive feature elimination (SVM-RFE) approach for extracting features while identifying cancer cells [33,34]. SVM-RFE is a sequential backward selection algorithm that is based on the SVM maximum interval principle. Consequently, the SVM-RFE ranking criteria are closely related to the SVM [35].

Give a training sample set $\{(x_i, y_i)\}_{i=1}^{N}, x_i \in R^D, y_i \in \{+1, -1\}$, where $y_i$ is the category label of $x_i$, N is the number of training samples, and D is the feature dimension of the samples. Furthermore, the SVM seeks the optimal classification plane $\omega x + b = 0$, where $\omega$ is the weight vector of the optimal hyperplane and $b$ is the threshold, so that the optimal classification plane not only separates the two classes of samples without error, but also maximizes the classification interval between the two classes.

In order to calculate the weight vector and threshold, the SVM needs to solve the following optimization problems:

$$\min \frac{1}{2}\|\omega\|^2 + C \sum_{i=1}^{N} \zeta_i \tag{1}$$

and

$$y_i(\omega \cdot x_i + b) \geq 1 + \varsigma_i; i = 1, 2, \cdots, N$$

$$\zeta_i \geq 0; i = 1, 2, \cdots, N$$

where $C > 0$ is the penalty parameter and $\zeta_i$ is the relaxation variable. The role of parameter $C$ is to adjust the level of penalty for sample misclassification and to achieve a trade-off between the percentage of sample misclassification and the algorithm's complexity.

By introducing Lagrange multipliers, the optimization problem of SVM can be transformed into the following pairwise programming problems:

$$\min \frac{1}{2} \sum_{i=1}^{N} \sum_{i=1}^{N} \alpha_i \alpha_j y_i y_j (x_i \cdot x_j) - \sum_{i=1}^{N} \alpha_i \tag{2}$$

and

$$\sum_{i=1}^{N} y_i \alpha_i = 0; 0 \leq \alpha_i \leq C, i = 1, 2, \cdots, N$$

where $\alpha_i$ is the Lagrange multiplier.

The relationship between the weight vector and the solution of the pairwise optimization (2) is:

$$\omega = \sum_{i=1}^{N} \alpha_i y_i x_i \tag{3}$$

In SVM-RFE, the ranking criterion score of the ith feature is defined as:

$$c_i = \omega_i^2 \tag{4}$$

where $\omega_i$ is the vector of weights of the optimal hyperplane.

Each round of recursive feature elimination needs training the SVM in order to obtain the scoring criterion. The feature with the lowest score is eliminated from the initial feature set in order to generate a new one, as it has the least impact on classification performance. The next iteration updates the feature set used to train the SVM. Repeat this procedure until all features have been removed, and then arrange the features in descending order of removal. The later eliminated elements are more important.

### 2.1.2. Feature Selection Using PSO-SVM

This paper uses binary particle swarm optimization (PSO) [36] as feature selection for classification problems. In each iteration, the particles are optimized according to

their fitness and swarm fitness values. Using SVM to evaluate the fitness value of particle swarm optimization and by introducing a kernel function, the maximum edge hyperplane suitable for the classification problem structure is found in the high feature space, thereby improving the efficiency of the fitness value function.

According to particle swarm optimization rules, we first set the required number of particles and then randomly generate an initial binary-coded string for each particle. For example, when using particle swarm optimization to analyze an eight-dimensional dataset $S_n = [H_1 H_2 H_3 H_4 H_5 H_6 H_7 H_8]$ ($n = 8$) to select features, we can select any number of features less than n. We can randomly select three features ($m = 3$), here $S_m = [H_1 H_5 H_7]$ When calculating the adaptive value, these m features in each dataset represent the data dimension d, which is evaluated by SVM. When the sample size is large, the fitness value of SVM evolves according to the Holdout method. Moreover, the kernel function of SVM is radial basis function (RBF):

$$K(x,y) = \exp\left(-\gamma \|x - y\|^2\right), y > 0 \tag{5}$$

For different classification problems, support vector machines need to set different parameters. $\gamma$ and $C$ are important. By properly adjusting these parameters, a better classification hyperplane can be obtained and the classification accuracy can be improved. This paper did not optimize the SVM parameter setting, but set the parameters as $\gamma = 2^0$ and $C = 2^{12}$ according to relevant literature. The optimization can be used as a direction for follow-up research.

Each particle update is based on its adaptive value. The fitness function designed in this paper is shown in Equation (6):

$$\text{fitness} = \omega_1 \times \text{accuracy}_{SVM} + \omega_2 \times \left(\sum f_i\right)^{-1} \tag{6}$$

where $\omega_1$ represents the weight of classification accuracy; accuracy $_{SVM}$ represents the classification accuracy of SVM; $\omega_2$ represents the weight of feature dimension; and $f_i$ represents the relative offset of the *i*-th feature dimension in the mask, where $f_i = 1$ means the feature is retained and $f_i = 0$ means the feature is filtered. The actual problem determines $\omega_1$ and $\omega_2$. In this paper, $\omega_1 = 0.2$, $\omega_2 = 0.8$.

The best fitness value updated by each particle is pbest, and the best fitness value in a group of pbest is gbest. Once we have pbest and gbest, we can track the location and velocity characteristics of pbest and gbest particles. Each particle is updated according to Equations (7) and (8):

$$v_{id}^{t+1} = \omega v_{id}^t + c_1 \times \text{rand}() \times \left(pbest_{id} - x_{id}^t\right) + c_2 \times \text{rand}() \times \left(gbest_{id} - x_{id}^t\right) \tag{7}$$

$$x_{id}^{t+1} = \begin{cases} 1 & \text{rand} < sigmoid\left(v_{id}^{t+1}\right) \\ 0 & \text{rand} \geq sigmoid\left(v_{id}^{t+1}\right) \end{cases} \tag{8}$$

The updated features of the velocity value $v_{id}^{t+1}$ are calculated by the function $sigmoid(v_{id}^{t+1})$. If $sigmoid(v_{id}^{t+1})$ is greater than the random number within (0,1), its position value $H_n(n = 1, 2, \ldots, m)$ is represented as 1, and this feature should be retained in the next iteration. If $sigmoid(v_{id}^{t+1})$ is less than $H_n(n = 1, 2, \ldots, m)$ is represented as 0, then this feature will not appear in the next iteration.

### 2.1.3. Two-Level Feature Selection Preprocessing Model

When PSO-SVM makes a "one-to-one" feature selection, a variable number of redundant features are filtered in each iteration, resulting in an unpredictable number of residual features. In addition, the selection of SVM parameters substantially affects classification accuracy, frequently resulting in "missing the mark." This study develops a two-level feature selection preprocessing model, PSO-SVM-RFE, to eliminate unanticipated mistakes

produced by parameters impacting the accuracy of feature selection. Initially, the original data features are filtered using PSO-SVM, and then the filtered features are further filtered by SVM-RFE to produce the final feature selection results. The process of this model is shown in Figure 1.

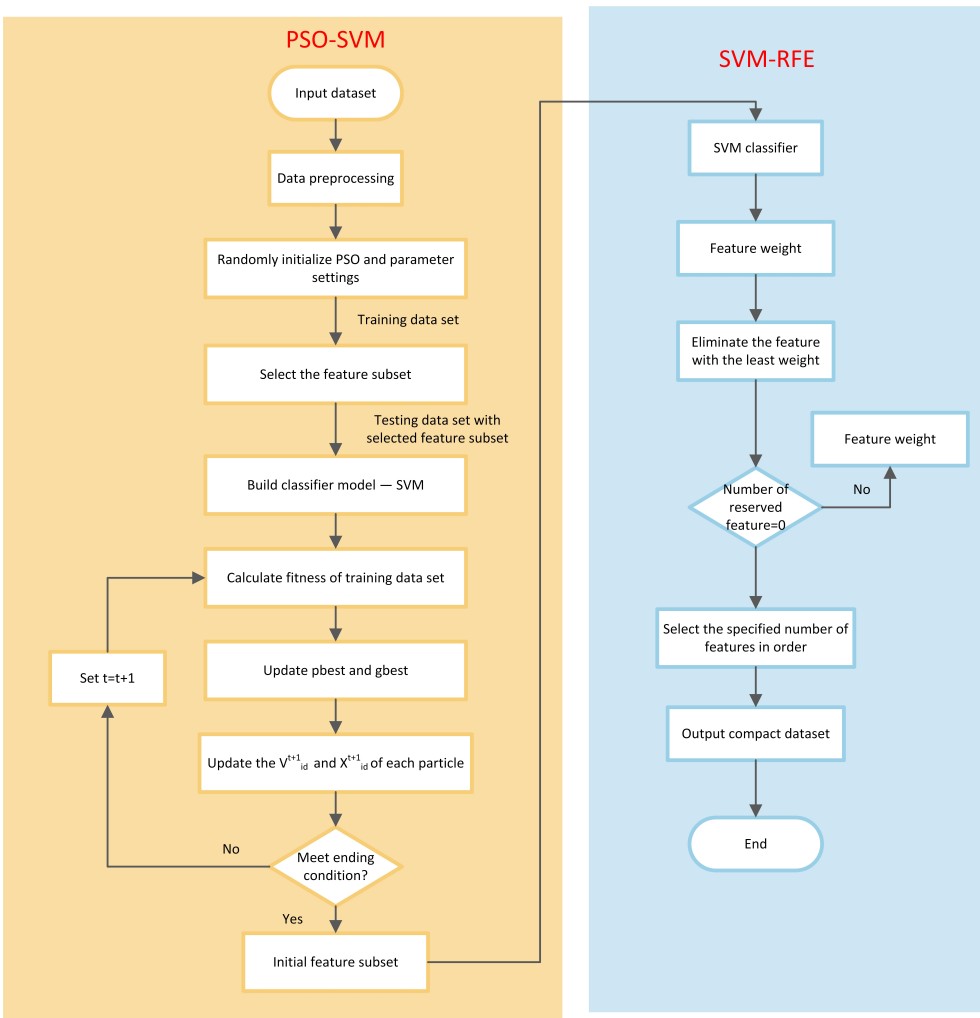

**Figure 1.** The proposed PSO-SVM-RFE method for feature selection.

### 2.2. Classification Phase

Artificial neural network (ANN) is a mathematical model that mimics the structure and function of biological neural networks and is used to estimate or approximate functions. Similar to other machine learning, neural networks have been used to solve various problems. However, ANN also has "fatal" shortcomings, which require some optimizers to make corresponding improvements. In this section, we first introduce the MSSA as an optimizer and then submit a variant of radial basis neural network, PNN, which is simple in structure and fast in training, especially suitable for pattern classification problems solution.

#### 2.2.1. Modified Salp Swarm Algorithm

Mirjalili et al. presented the salp swarm algorithm (SSA) in 2017 as a heuristic algorithm based on mimicking the group behaviour of salps in nature. SSA developed a salp chain model for solving optimization issues and separated salp groups into the leader and follower categories. The leader is the individual at the front of the chain, while the remaining individuals follow each other directly or indirectly as followers. SSA employs the approach of survival of the fittest. SSA continues to approach the food source position by calculating all individuals' adaptive values and comparing the current iterations' adaptive

values to the previous optimal ones. Thus, it is possible to model the foraging behaviour of salps to address the optimization problem [37–39].

Similar to other meta-heuristic algorithms, the original SSA had flaws, such as a sluggish convergence rate and a tendency to reach a local optimum during the optimization procedure quickly. This study offers a salp swarm method employing the Lévy flight strategy for the conditional update as a response.

Paul Lévy, a French mathematician, proposed the Lévy flight [40,41]. It is a distinct random walk approach. In the walking process, Lévy flight is accompanied by frequent short trips and occasional big distances, successfully balancing local development and global exploration capacity.

The random step size of Lévy's flight obeys Lévy's distribution, and its simplified form is:

$$\text{Levy}(s) = |s|^{-1-\beta}, 0 < \beta < 2 \tag{9}$$

where $s$ is the random step size. Since Lévy flight is very complex, this paper adopts the algorithm proposed by Mantegna to calculate [42], and its equation is as follows:

$$s = \frac{u}{|v|^{1/\beta}} \tag{10}$$

where $u$ and $v$ is a random number that is normally distributed, $u \sim N(0, \sigma_u^2), vs. \sim N(0, \sigma_v^2)$. $\sigma_u$ and $\sigma_v$ can be obtained from Equation (11):

$$\begin{cases} \sigma_u = \left\{ \frac{\Gamma(1+\beta) \cdot \sin(\pi\beta/2)}{\Gamma[(1+\beta)/2] \cdot \beta \cdot 2^{(\beta-1)/2}} \right\}^{1/\beta} \\ \sigma_v = 1 \end{cases} \tag{11}$$

where $\Gamma$ is the integral operation and $\beta$ usually takes the value of 1.5.

The mathematical algorithm of MSSA is described in detail below.

- **Step 1:** Initialization phase

At this stage, MSSA generates scattered initial random locations based on the size of the input dataset.

The target environment is defined as an $N \times D$ dimensional space, where $N$ represents the number of populations and $D$ represents the dimension of the space. The location of each salp is defined as $X_i = [X_{i1}, X_{i2}, X_{i3}, \cdots, X_{iD}], i = 1, 2, 3 \cdots, N$, and the target location is defined as $F = [F_1, F_2, F_3, \cdots, F_D]$. The upper bound of the search range of each dimension is $Ub = [ub_1, ub_2, ub_3, \cdots, ub_D]$, and the lower bound is $Lb = [lb_1, lb_2, lb_3, \cdots, lb_D]$. Finally, the initial population position is randomly obtained according to Equation (12):

$$X_{N_X D} = \text{rand}(N, D) \times (Ub - Lb) + Lb \tag{12}$$

In the population, the value of each dimension of the leader is defined as $X_d^1$, and the value of each dimension of the follower is defined as $X_d^n$, where $n = 2, 3, 4, \ldots, N$, $d$ represents the dimension.

- **Step 2:** Improve update strategy for leader positions

Lévy's random flight step was used to improve the position update of the leader. Lévy flight strategy enables the algorithm to change randomly between long and short distances, and a small number of long hops are used to avoid the algorithm falling into local optimization to enhance the global search ability. Leaders update positions according to Equation (13):

$$X_d^1 = \begin{cases} F_d + c_1((ub - lb) \oplus \text{Levy}(\lambda) + lb), c_3 \geq 0.5 \\ F_d - c_1((ub - lb) \oplus \text{Levy}(\lambda) + lb), c_3 < 0.5 \end{cases} \tag{13}$$

where $F_d$ is the base target position, $\text{Levy}(\lambda)$ is the Lévy flight path, $c_1$ and $c_3$ are control parameters, $c_3$ is a random number between [0,1], which determines the direction and

step size of the leader's position update, and $c_1$ is the convergence factor, which is used to balance the convergence speed of the algorithm in the iterative process, as shown in Equation (14):

$$c_1 = 2e^{-(4 \times l/l_{\max})^2} \tag{14}$$

where $l$ is the number of iterations and $l_{max}$ is the maximum number of iterations.

- **Step 3:** Improve update strategy for follower positions

In the original SSA, the follower blindly followed the previous salp, making it miss the better fitness position. In the improved algorithm MSSA, a conditional "piecewise" location update is adopted: firstly, the fitness of the previous salp is compared with the current fitness to select the new location so that the new location is more inclined to the side with better fitness. Compared with blind random updating, this mechanism can approach the optimal solution faster, thus speeding up the convergence speed of the algorithm. Therefore, the position of followers is updated in the improved way shown in Equation (15):

$$X_d^n = \varepsilon \left( X_d^{n-1} + X_d^n \right) \tag{15}$$

where $\varepsilon$ is the coefficient of position offset, and its calculation equation is as follows:

$$\varepsilon = \begin{cases} 0.5 \times \text{rand}(0,1) & f\left(X_d^{n-1}\right) < f\left(X_d^n\right) \\ 0.5 & f\left(X_d^{n-1}\right) = f\left(X_d^n\right) \\ 1 - 0.5 \times \text{rand}(0,1) & f\left(X_d^{n-1}\right) > f\left(X_d^n\right) \end{cases} \tag{16}$$

where $f$ is the fitness function.

According to the update method of population individuals described above, MSSA can be obtained, as shown in Algorithm 1.

---

**Algorithm 1** Modified salp swarm algorithm (Pseudo-code)

---

1: Initialization parameters: population size $N$, dimension $D$, maximun number of iterations $l_{\max}$ .
2: Generate the initial population $X_{N \times D}$ by Equation (11);
3: Calculate the fintess value for each individual search agent.
4: While $l \leq l_{\max} + 1$ do
5:     Update $c_1$ by Equation (13);
6:     for $i = 1 : N$ do
7:         if $X_{i=1}(\text{leader})$ then
8:             Update random numbers $c_3$ and $\beta$;
9:             Update the position of the leader salp as in Equation (19);
10:         else
11:         Update $\varepsilon$ by Equation (16);
12:         Update the position of the follower salp as in Equation (20);
13:         end if
14:     end for
15:     set $l = l + 1$;
16: end while
Output: Best classification and predication results.

---

### 2.2.2. Probabilistic Neural Network

The probabilistic neural network (PNN) is a radial basis function network based on the Bayesian decision theory. PNN features easy training, fast convergence, and arbitrary nonlinear methods. Due to the specificity of the functions it relies on, PNN is highly fault-tolerant and robust.

PNN comprises the input, pattern, summation, and output layers. Nodes in the input layer are a set of predicted values. The pattern layer consists of Gaussian functions

centred on the prediction set. The summation layer averages each set of predicted values. The output layer determines the class label associated with it by voting. Figure 2 shows the structure of the PNN.

For the input vector, to match various features of the training set, each unit output is as follows:

$$\phi_{ij}(x) = \frac{1}{2\pi^{\frac{d}{2}}\sigma^d} \exp\left(-\frac{(X - x_{ij})^T (X - x_{ij})}{\sigma^2}\right) \tag{17}$$

where $X = [x_1, x_2, x_3, \ldots, x_n]^T$, $n = 1, 2, \ldots, l$; $d$ is the dimension of the feature vector; $l$ is all training types; $x_{ij}$ is the jth centre of the ith training sample; and $\sigma$ is the smooth factor.

The output weights of summation layer neurons are calculated as follows:

$$v_i = \frac{\sum_{j=1}^{L} \phi_{ij}}{L} \tag{18}$$

where $v_i$ is output of ith type and $L$ is the number of class ith neurons.

The output layer takes the type corresponding to the maximum output weight obtained by the summing layer as the output type, and the result is as follows:

$$\text{Type}(v_i) = \arg\max(v_i) \tag{19}$$

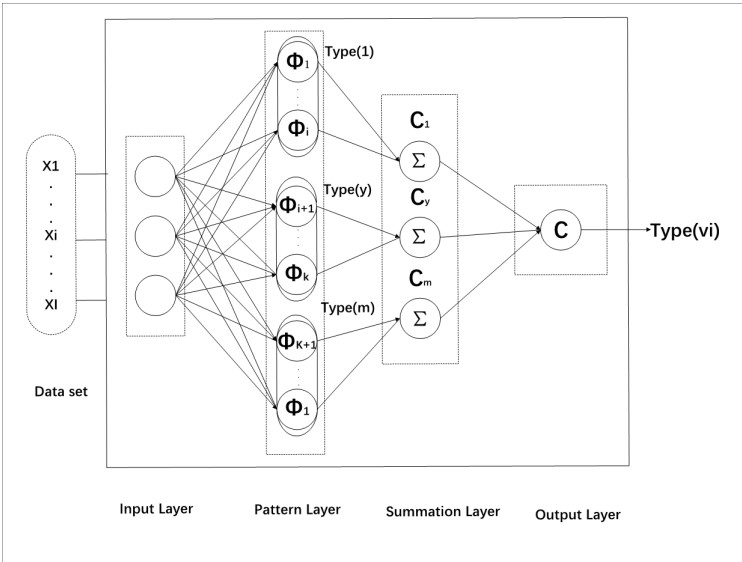

**Figure 2.** The structure of PNN.

### 2.3. The Proposed PSO-SVM-RFE-MSSA-PNN Model

The PNN structure diagram reveals that the smoothing factor affects the classification performance of PNN straightforwardly. If it is too large or too little, the convergence of the network will be too quick or too slow, preventing the optimal solution from being identified. As a result, the diagnostic accuracy and classification performance of the PNN will be drastically diminished. Since MSSA has significant benefits over other optimization algorithms in terms of global search and population diversity, this research employs MSSA to improve the classification performance of PNN by locating an appropriate one. To save storage capacity and improve the accuracy of diagnosis, we use PSO-SVM-RFE for feature selection. It eliminates redundant features and minimizes the input's feature dimension, thus obtaining a highly representative feature volume for various fault types. The design process of PSO-SVM-RFE-MSSA-PNN is shown in Figure 3, and the specific steps are as follows:

Step 1:  The data samples were entered into PSO-SVM-RFE and ranked in order of importance for each feature.

Step 2: Select the specified number of features to construct feature subsets based on the ranking results, and obtain the simplified sample dataset based on the feature subsets.

Step 3: The simplified data samples are preprocessed and then randomly input to PNN.

Step 4: The initial parameters of MSSA are set as follows: the number of populations N, dimension d, and the maximum number of iterations $l_{max}$. In addition, the population positions of MSSA are initialized by Equation (12).

Step 5: The fitness of salp individuals in the initial population is calculated and ranked. The fitness function in this paper is set as the mean square error function, as shown in Equation (20):

$$f(x) = \sqrt{\frac{1}{n}\sum_{i=1}^{N}(Y_i - O_i)^2} \tag{20}$$

where $Y_i$ and $O_i$ are the training accuracy and testing accuracy under the effect of a particular smoothing factor, respectively.

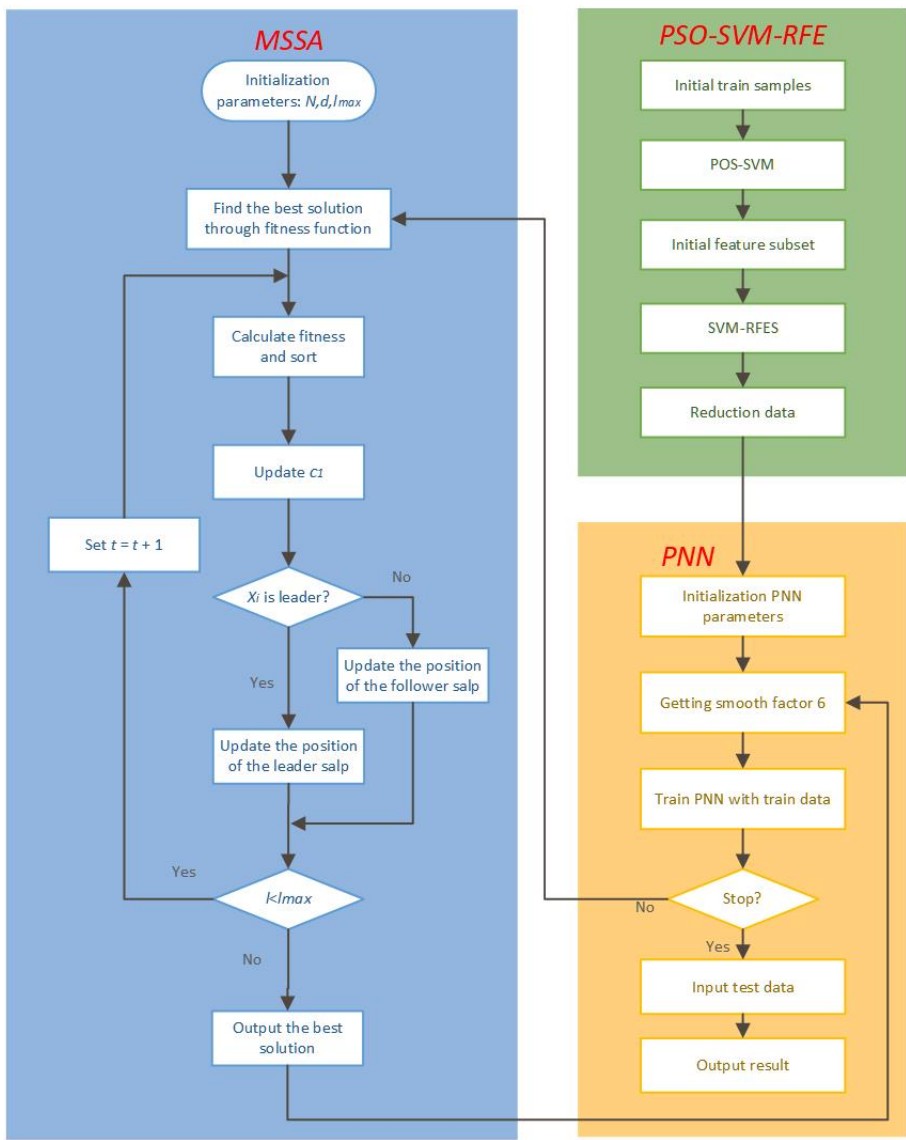

**Figure 3.** The proposed PSO-SVM-RFE-MSSA-PNN method for fault diagnosis.

Step 6: The salp individual location with the best fitness was considered to be the current food location. Of the remaining N-1 salp individuals, the most adaptable salp is considered the leader, and the rest are considered followers.

Step 7: Update $c_1$ according to Equation (14).
Step 8: Update the leader position by Equation (13) and the follower position by Equation (15).
Step 9: The following process is continued if the maximum number of iterations is reached or the preset conditions are met. If not, return to step 6.
Step 10: At the end of the training process, the optimized smoothing factor $\sigma$ is input into PNN to obtain a PNN with global optimization performance. Then, input the test sample data into PNN to obtain the final diagnosis result.

## 3. Results and Discussions

### 3.1. Tennessee Eastman Process

An American chemical corporation established the TE process in 1993 as a chemical modeling and simulation platform. The TE process is a classic chemical process, commonly employed for process monitoring and problem diagnosis [43–46]. Figure 4 depicts an approximation of the TE process flowchart. There are four gaseous reactants in the TE process, A, D, C, and E, and two products, G and H.

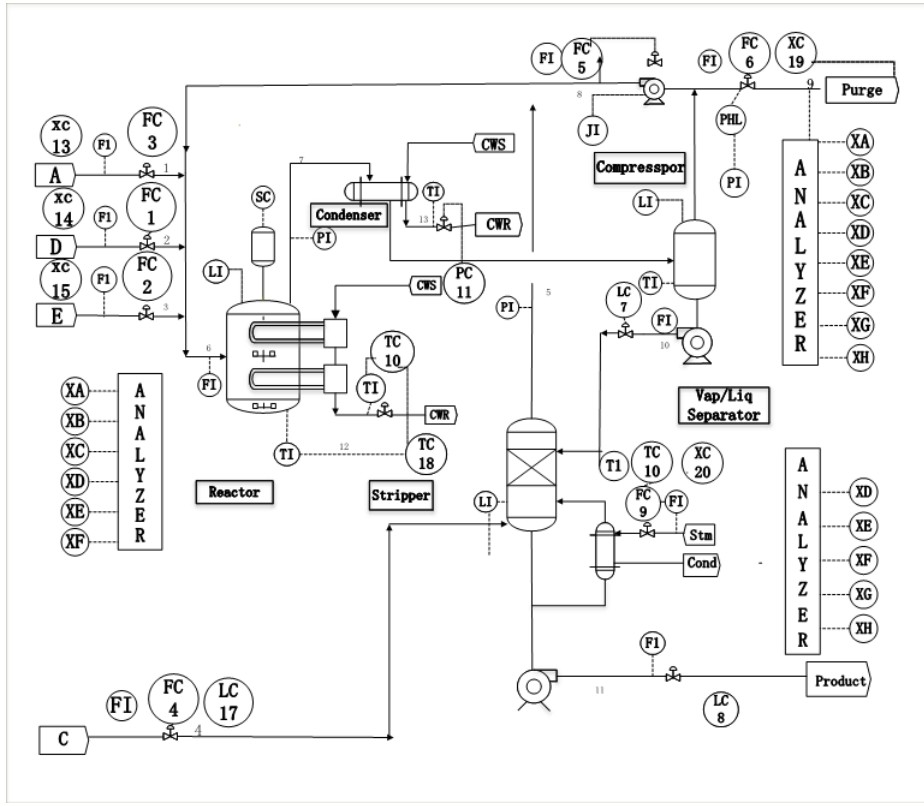

**Figure 4.** The Tennessee Eastman process diagram.

Table 1 displays the defect categories in the TE process database. Fault categories 1–7 are order variable faults, 8–12 are random variable faults, 13 are slow drift faults in chemical reaction dynamics, 14–15 are corresponding viscous faults, 16–20 are unknown faults, and 21 are constant position faults. The TE process contains 41 measurements and 11 control variables, and Table 2 lists all the variables associated with this process.

The TE process database includes the following data sources. It has been set to three minutes for the sampling intervals. After 48 h of continuous operation under normal process conditions, 960 samples were obtained as standard data samples. After 8 h of normal operation, 21 defects were introduced, which lasted for 48 h until the chemical process was completed. As a result, of the 960 data samples collected during a failure, the first 160 were captured during normal operation, and the remaining 800 were collected

after the failure. If the reader is interested in learning more about these datasets, they can be found at the following address: http://web.mit.edu/braatzgroup/links.html, accessed on 25 June 2022.

**Table 1.** Description of fault categories.

| Category | Description | Type |
|---|---|---|
| 1 | A/C Feed ratio, B composition constant (Stream 4) | Step |
| 2 | B composition, A/C ratio constant (Stream 4) | Step |
| 3 | D feed temperature (Stream 2) | Step |
| 4 | Reactor cooling water inlet temperature | Step |
| 5 | Condenser cooling water inlet temperature | Step |
| 6 | A feed loss (Stream 1) | Step |
| 7 | C header pressure loss (Stream 4) | Step |
| 8 | A, B, C, feed composition (Stream 4) | Random variation |
| 9 | D feed temperature (Stream 2) | Random variation |
| 10 | C feed temperature (Stream 4) | Random variation |
| 11 | Reactor cooling water inlet temperature | Random variation |
| 12 | Condenser cooling water inlet temperature | Random variation |
| 13 | Reaction kinetics | Slow drift |
| 14 | Reactor cooling water valve | Sticking |
| 15 | Condenser cooling water valve | Sticking |
| 16–20 | Unknown | Unknown |
| 21 | Valve (Stream 4) | Constant position |

**Table 2.** Measured and manipulated variables.

| NO. | Process Measurements | NO. | Process Measurements |
|---|---|---|---|
| 1 | A feed | 27 | E in reactor feed |
| 2 | D feed | 28 | F in reactor feed |
| 3 | E feed | 29 | A in reactor feed |
| 4 | Total feed | 30 | B in reactor feed |
| 5 | Recycle flow | 31 | C in reactor feed |
| 6 | Reactor feed rate | 32 | D in reactor feed |
| 7 | Reactor pressure | 33 | E in reactor feed |
| 8 | Reactor level | 34 | F in reactor feed |
| 9 | Reactor temperature | 35 | G in reactor feed |
| 10 | Purge rate | 36 | H in reactor feed |
| 11 | Product separator temperature | 37 | D in product flow |
| 12 | Product separator level | 38 | E in product flow |
| 13 | Product separator pressure | 39 | F in product flow |
| 14 | Product separator underflow | 40 | G in product flow |
| 15 | Stripper level | 41 | H in product flow |
| 16 | Stripper pressure | 42 | D feed flow valve |
| 17 | Stripper underflow | 43 | E feed flow valve |
| 18 | Stripper temperature | 44 | A feed flow valve |
| 19 | Stripper steam flow | 45 | Total feed flow valve |
| 20 | Compressor work | 46 | Compressor recycle valve |
| 21 | Reactor cooling water outlet temperature | 47 | Purge valve |
| 22 | Separator cooling water outlet temperature | 48 | Separator pot liquid flow valve |
| 23 | A in reactor is feed | 49 | Stripper liquid product flow valve |
| 24 | B in reactor is feed | 50 | Stripper steam valve |
| 25 | C in reactor is feed | 51 | Reactor cooling water flow |
| 26 | D in reactor is feed | 52 | Condenser cooling water flow |

*3.2. Three Experiments to Verify the Validity of the Proposed Model*

This paper examines the performance of the PSO-SVM-RFE-MSSA-PNN model from three different angles. First, various feature selection approaches have varying effects on simplifying high-dimensional data into low-dimensional features significantly connected

with fault categories. Second, different optimization techniques use different updating strategies for PNN smoothing factors. Third, since different classifiers use different classification rules, the diagnosis results will also differ. As a result, the following three separate experiments are defined by the above viewpoint. The MATLAB software environment was used for the above studies.

### 3.2.1. Influence of Different Feature Selection Algorithms on the Performance of Fault Diagnosis

TE datasets are high-dimensional, small imbalanced samples, and three basic feature selection methods exist filter, wrapper, and embedding. SVM-RFE, an embedded feature selection method, was utilized by Yang X et al. [47] to score and sort the features, and the best five features were then selected for classification studies. Table 3 shows the feature selection results. Xie Z et al. [48] utilized the filtered feature selection method random forest three-bagger (RFtb), randomly divided the dataset, incrementally grew the decision tree set from the given dataset, and measured and sorted the relative value of 52 features. The initial five characteristics are chosen for categorization experiments. The results of the feature selection are shown in Table 4. This study offers a packaged feature selection approach, PSO-SVM-RFE, which employs PSO-SVM and SVM-RFE to analyze the original dataset and produce a dataset with five features. The results of the feature selection are shown in Table 5.

**Table 3.** Feature selection results of SVM-RFE.

| Category | Feature | Category | Feature | Category | Feature |
|---|---|---|---|---|---|
| 1 | 18,16,7,46,44 | 8 | 50,19,18,13,16 | 15 | 17,52,18,7,20 |
| 2 | 10,7,47,20,19 | 9 | 52,17,13,7,19 | 16 | 17,52,48,12,7 |
| 3 | 52,17,11,19,18 | 10 | 13,7,50,19,18 | 17 | 21,7,13,9,51 |
| 4 | 51,9,21,18,19 | 11 | 52,17,48,12,16 | 18 | 52,17,50,18,20 |
| 5 | 52,17,11,19,18 | 12 | 7,13,50,19,16 | 19 | 52,17,48,12,20 |
| 6 | 1,44,36,26,10 | 13 | 50,19,13,52,17 | 20 | 52,17,13,7,20 |
| 7 | 45,7,35,25,16 | 14 | 52,17,51,9,13 | 21 | 17,52,19,18,50 |

**Table 4.** Feature selection results of RFtb.

| Category | Feature | Category | Feature | Category | Feature |
|---|---|---|---|---|---|
| 1 | 1,20,22,44,46 | 8 | 16,29,38,40,41 | 15 | 16,19,20,39,40 |
| 2 | 10,34,39,46,47 | 9 | 19,25,31,38,50 | 16 | 18,19,38,46,50 |
| 3 | 18,20,37,40,41 | 10 | 18,19,31,38,50 | 17 | 21,38,46,50,51 |
| 4 | 19,38,47,50,51 | 11 | 7,9,13,38,51 | 18 | 16,19,22,41,50 |
| 5 | 17,18,38,50,52 | 12 | 4,11,18,19,35 | 19 | 5,13,20,46,50 |
| 6 | 1,20,37,44,46 | 13 | 7,18,19,39,50 | 20 | 19,39,41,46,50 |
| 7 | 19,38,45,46,50 | 14 | 9,11,21,38,50 | 21 | 7,16,19,45,50 |

**Table 5.** Feature selection results of PSO-SVM-RFE.

| Category | Feature | Category | Feature | Category | Feature |
|---|---|---|---|---|---|
| 1 | 40,42,13,44,18 | 8 | 17,24,28,34,52 | 15 | 28,29,34,39,35 |
| 2 | 5,29,40,47,51 | 9 | 28,29,34,38,41 | 16 | 24,26,29,33,40 |
| 3 | 19,18,37,41,40 | 10 | 25,28,29,34,35 | 17 | 7,50,20,23,38 |
| 4 | 29,35,40,42,51 | 11 | 19,7,27,39,16 | 18 | 4,17,22,32,52 |
| 5 | 8,17,22,35,52 | 12 | 3,44,13,20,47 | 19 | 18,25,29,32,35 |
| 6 | 46,26,13,16,20 | 13 | 13,16,24,50,41 | 20 | 7,37,14,41,35 |
| 7 | 7,16,20,31,45 | 14 | 21,51,43,29,25 | 21 | 11,18,35,37,50 |

First, the training samples under normal state and 21 fault categories are fed into the PSO-SVM-RFE model for feature selection, and the five highest-priority feature ranking sets are produced. The training and test samples are then simplified using our feature selection



sort set and Yang X and Xie Z's feature selection sort sets. Lastly, the three simplified samples and the original samples without simplification are entered into the optimization model MSSA-PNN established in this research to compare the performance of various feature selection approaches. The population of the model is set at 20, and the maximum number of iterations is 30.

Feature selection techniques such as PSO-SVM-RFE and others are shown in Table 6. As a result of the optimization, the diagnostic rate for the 21 faults was 91% for PSO-SVM-RFE, 88.8% for SVM-RFE, and 90% with RFtb. After feature selection with PSO-SVM-RFE, the smaller dataset is more accurate because of the following.

①  The PSO-SVM-RFE model offers a considerable advantage in feature selection for categories 3, 9, 11, 13, 14, 17, and 19, compared to the SVM-RFE model for these seven fault categories. Despite the fact that the PSO-SVM-RFE model does not outperform the SVM model in categories 8, 10, 15, and 16, its diagnosis rate is still higher than 80%. In the remaining categories of defects, the differences between them are insignificant. Thus, when it comes to selecting features, the PSO-SVM-RFE model with initial pre-screening by PSO-SVM does better than the SVM-RFE model.

②  For categories 3, 9, 11, 13, 14, 17, and 19, the PSO-SVM-RFE model has a significant advantage over the SVM-RFE model in feature selection for these seven fault categories. Although the PSO-SVM-RFE model does not have an advantage over the SVM-RFE model for categories 8, 10, 15, and 16, its diagnostic rate is still above 80.00%. In the other fault categories, the difference between them is not significant. Thus, the PSO-SVM-RFE model with initial pre-screening by PSO-SVM performs better in feature selection than the single SVM-RFE model.

③  PSO-SVM-RFE had an average diagnostic rate of 91%, whereas RFtb had an average diagnostic rate of 90%. The former has a minor edge in the average accuracy rate if there is little difference in the diagnostic rate of 21 fault categories. PSO-SVM-RFE has a better diagnostic rate for several fault types than RFtb. Figure 5 shows that the PSO-SVM-RFE algorithm gives a better diagnosis rate for the 21 fault categories while keeping a more or less smooth quasi-break rate.

**Table 6.** The fault diagnosis rates of various feature selection models.

| Category | PSO-SVM-RFE | SVM-RFE | RFtb | Unoptimized |
|---|---|---|---|---|
| 1 | 0.99 | 1.00 | 0.93 | 0.99 |
| 2 | 0.99 | 0.99 | 0.99 | 0.99 |
| 3 | 0.98 | 0.69 | 0.98 | 0.85 |
| 4 | 1.00 | 1.00 | 1.00 | 1.00 |
| 5 | 1.00 | 1.00 | 0.93 | 0.83 |
| 6 | 0.99 | 1.00 | 1.00 | 0.99 |
| 7 | 1.00 | 1.00 | 1.00 | 1.00 |
| 8 | 0.83 | 0.96 | 0.99 | 0.91 |
| 9 | 0.87 | 0.71 | 0.82 | 0.84 |
| 10 | 0.82 | 0.91 | 0.96 | 0.77 |
| 11 | 0.84 | 0.64 | 0.91 | 0.75 |
| 12 | 0.94 | 0.95 | 0.97 | 0.90 |
| 13 | 0.98 | 0.87 | 1.00 | 0.94 |
| 14 | 1.00 | 0.92 | 1.00 | 0.83 |
| 15 | 0.82 | 0.94 | 0.76 | 0.80 |
| 16 | 0.86 | 0.93 | 0.62 | 0.96 |
| 17 | 0.87 | 0.73 | 0.70 | 0.86 |
| 18 | 0.88 | 0.86 | 0.95 | 0.86 |
| 19 | 0.86 | 0.71 | 0.94 | 0.77 |
| 20 | 0.80 | 0.84 | 0.85 | 0.84 |
| 21 | 0.84 | 0.79 | 0.54 | 0.82 |
| Mean | 0.91 | 0.88 | 0.90 | 0.88 |

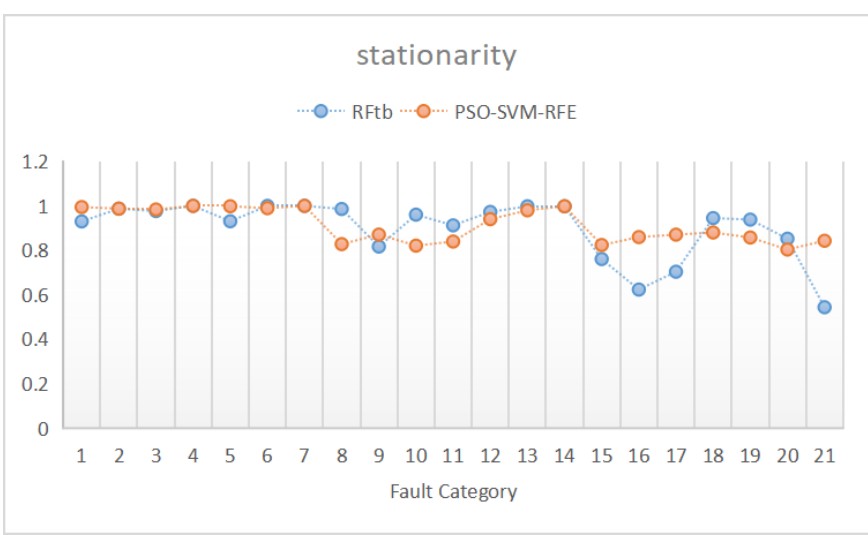

**Figure 5.** Stability comparison of PSO-SVM-RFE and RFtb for 21 fault diagnosis rates.

### 3.2.2. Performance Analysis of Fault Classification for Different Optimized PNN Schemes

To compare the optimization performance of various heuristic methods on the smoothing factor of PNN, the training sample set and the test sample set are generated using the simplified feature set resulting from PSO-SVM-RFE feature selection, as shown in Table 5. The training set is then inputted into the MSSA-PNN, SSA-PNN, genetic algorithm (GA)-PNN, cuckoo algorithm (CS)-PNN, PSO-PNN, seagull optimization algorithm (SOA)-PNN, multi-verse optimizer (MVO)-PNN, and Unoptimized PNN models. Lastly, the test set is utilized to compare the respective fault correctness rates. All models have 20 populations and 50 iterations. GA-PNN sets the crossover probability to 0.7 and the mutation probability to 0.01. The CS-PNN sets the discovery probability Pa to 0.25, $\lambda$ to 1, and the step size $\alpha$ to 0.4. PSO sets the maximum speed to 1, the minimum speed to 1, the solution space to $[-5, 5]$, and the learning factor to 1.49445 [48]. The smoothing factor of the Unoptimized PNN is 0.8.

The results of the classification are provided in Table 7. As shown in the table, the average MSSA-PNN diagnostic rate is 88%, which is more than other approaches: 77% for the PNN, 86% for the SSA-PNN, 83% for the CS-PNN, 81% for the GA-PNN, 84% for the PSO-PNN, 82% for the MOV-PNN, and 86% for the SOA-PNN. The results are analyzed as follows.

① Compared to the other optimization models, the diagnostic rate of PNN is lower on average, which demonstrates the inadequacies of the common PNN and the importance of the optimization model.

② MSSA-PNN has a higher average diagnostic rate than SSA-PNN, demonstrating the advantages of optimized SSA and the soundness of this theory.

③ MSSA-PNN has the highest average diagnosis rate among the aforementioned optimization models, and its fault diagnosis rate is superior than those of other optimization models. In addition, categories 3, 9, 13, and 17 outperform all other categories. Although it is inferior in a few defect categories, the difference is minor. Further evidence demonstrates that utilizing MSSA to optimize PNN can boost performance.

"G at reactor feed" (characteristic 35) can indicate product quality. Failure categories 1, 2, 5, 6, 7, 8, 12, 13, 20, and 21 have a stronger influence on product quality than other failure categories [48]. The actual TE process is imitated, i.e., the actual situation is reproduced when the chemical process fails under normal circumstances. In this paper, 270 random samples are taken from the original dataset, and 570 random samples are taken from 12(13) categories of faults to make a "simulated sample set." The chemical process with this set of samples is as follows: the sampling interval is 3 minutes, and the TE process is carried out normally when $t \in (0, 3 \times 270)$ min; the TE process is carried out under normal

conditions. $t = (3 \times 270 + 1)$ min marked the beginning of the category 12(13) fault, which persisted until $t = 3 \times (270 + 570)$ min was reached.

**Table 7.** Troubleshooting rate of different optimized PNN schemes.

| Category | MSSA-PNN | SSA-PNN | CS-PNN | GA-PNN | PSO-PNN | MOV-PNN | SOA-PNN | PNN |
|---|---|---|---|---|---|---|---|---|
| 1 | 1.00 | 1.00 | 1.00 | 0.99 | 0.99 | 0.99 | 0.96 | 0.99 |
| 2 | 0.98 | 0.98 | 0.98 | 0.98 | 0.98 | 0.96 | 0.97 | 0.97 |
| 3 | 0.98 | 0.94 | 0.70 | 0.70 | 0.71 | 0.72 | 0.93 | 0.66 |
| 4 | 1.00 | 1.00 | 1.00 | 1.00 | 1.00 | 0.99 | 1.00 | 0.97 |
| 5 | 1.00 | 1.00 | 0.99 | 0.99 | 0.99 | 1.00 | 0.99 | 0.67 |
| 6 | 0.99 | 0.94 | 0.98 | 0.99 | 0.89 | 0.98 | 0.90 | 0.95 |
| 7 | 1.00 | 1.00 | 1.00 | 0.97 | 1.00 | 0.66 | 1.00 | 1.00 |
| 8 | 0.80 | 0.80 | 0.82 | 0.83 | 0.84 | 0.78 | 0.80 | 0.67 |
| 9 | 0.82 | 0.79 | 0.75 | 0.82 | 0.77 | 0.70 | 0.79 | 0.65 |
| 10 | 0.79 | 0.73 | 0.81 | 0.81 | 0.82 | 0.67 | 0.73 | 0.67 |
| 11 | 0.78 | 0.79 | 0.69 | 0.63 | 0.67 | 0.74 | 0.74 | 0.67 |
| 12 | 0.88 | 0.84 | 0.59 | 0.35 | 0.94 | 0.91 | 0.76 | 0.66 |
| 13 | 0.98 | 0.96 | 0.89 | 0.76 | 0.79 | 0.90 | 0.95 | 0.82 |
| 14 | 1.00 | 1.00 | 1.00 | 1.00 | 1.00 | 1.00 | 0.99 | 1.00 |
| 15 | 0.76 | 0.76 | 0.80 | 0.72 | 0.84 | 0.72 | 0.78 | 0.66 |
| 16 | 0.78 | 0.74 | 0.80 | 0.79 | 0.80 | 0.74 | 0.69 | 0.66 |
| 17 | 0.84 | 0.83 | 0.59 | 0.72 | 0.73 | 0.75 | 0.84 | 0.66 |
| 18 | 0.84 | 0.73 | 0.83 | 0.86 | 0.82 | 0.82 | 0.85 | 0.85 |
| 19 | 0.79 | 0.77 | 0.79 | 0.76 | 0.67 | 0.72 | 0.80 | 0.65 |
| 20 | 0.71 | 0.69 | 0.62 | 0.64 | 0.64 | 0.68 | 0.71 | 0.66 |
| 21 | 0.81 | 0.83 | 0.80 | 0.79 | 0.75 | 0.83 | 0.82 | 0.66 |
| Mean | 0.88 | 0.86 | 0.83 | 0.81 | 0.84 | 0.82 | 0.86 | 0.77 |

Figures 6 and 7 show detailed process monitoring graphs for two typical faults (category 12 and category 13). These graphs combine the raw time trends with the dynamics of the TE process. Because of this, it is easier to see how well the various optimization models monitor the dynamics of the TE process for the category 12(13) failure, which has a significant impact on the product quality, and it is possible to visualize the data in Table 7, making the "numerical" experimental results more informative.

For category 12, Figure 6 shows that the original PNN has an overfitting phenomenon in diagnosing category 12 faults and failing to separate the normal state data. The GA-PNN also suffers from overfitting when optimizing the smoothing factor but behaves exactly opposite to the original PNN, failing to isolate the fault state data. SSA-PNN and SOA-PNN do not identify the normal state data very well, and the CS-PNN does not identify the fault state data very well. MSSA-PNNN, PSO-PNN, and MOV-PNN are generally close to each other in terms of fault detection performance. However, a closer look reveals that MSSA-PNN is more sensitive than PSO-PNN for fault state data and more sensitive than MOV-PNN for normal state data.

Similar to category 12, category 13 has a detrimental effect on the overall quality of the product. As can be seen in Figure 7, the CS-PNN, GA-PNN, PSO-PNN, MOV-PNN, and the original PNN fail to achieve the desired results for monitoring category 13 faults. The fault detection capabilities of the MSSA-PNN, SSA-PNN, and SOA-PNN are generally equal. However, MSSA-PNN and SSA-PNN have a slightly higher recognition rate for fault state data than SSA-PNN, with almost the same recognition rate for normal state data. Although SOA-PNN has a slightly higher recognition rate for normal state data than MSSA-PNN, its recognition rate for fault state data is significantly lower than that of MSSA-PNNN.

In summary, MSSA-PNN has high robustness and good diagnostic accuracy. Moreover, it is more sensitive than other optimization models and can accurately classify fault and non-fault data. It is also not prone to overfitting problems that degrade the model's performance. Therefore, on the whole, the fault detection performance of MSSA-PNN is superior.

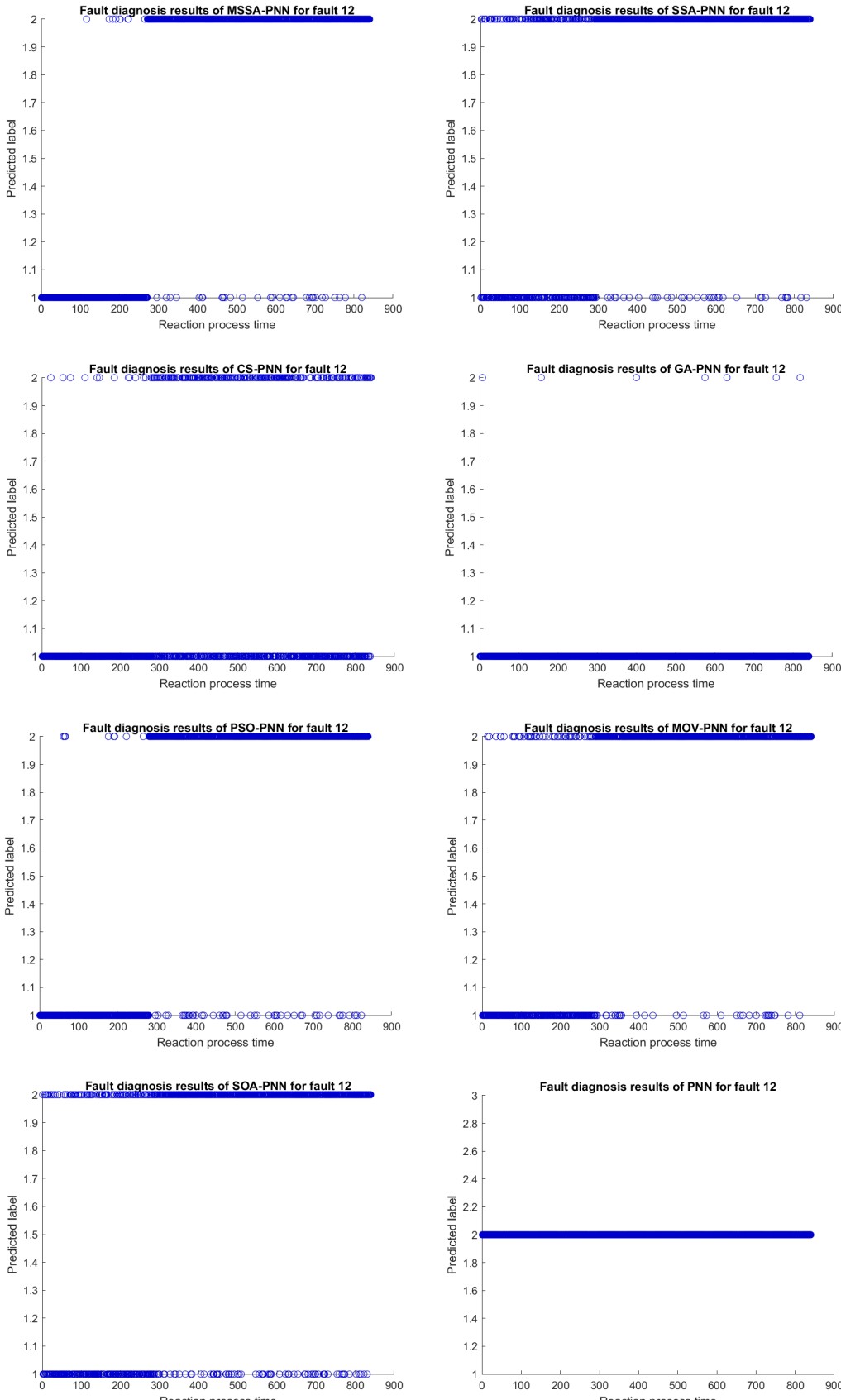

**Figure 6.** Monitoring performances for Fault 12.

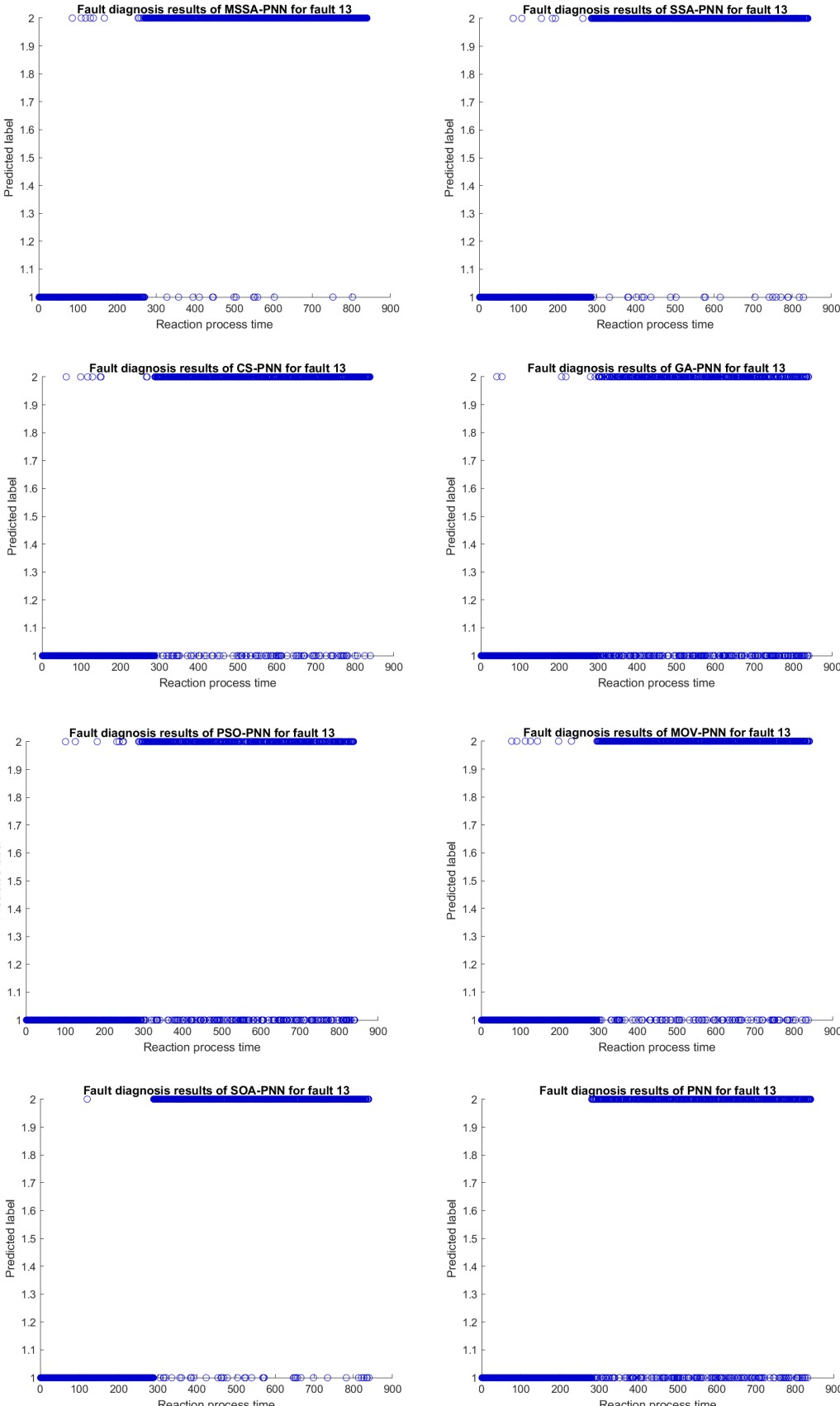

**Figure 7.** Monitoring performances for Fault 13.

### 3.2.3. Analysis of Fault Diagnosis Performance Indicators of Different Classifiers

In the third experiment, the performance of the MSSA-PNN classification model was evaluated by diagnosing 21 types of errors in the TE process. The matching simplified data samples were generated based on the feature selection sorting set provided in Table 5 and then fed into the MSSA-PNN model and other commonly used classifiers to compare the classification outcomes. Similar to linear discriminant analysis (LDA) are quadratic discriminant analysis (QDA), KNN, SVM, and maximum entropy model (MaxEnt), which are used a lot in the literature [49–53]. In addition, we compare it to the hybrid back propagation (BP) neural network model described in the literature, such as (CS)-BP [54]. Moreover, the MSSA algorithm's parameters are consistent with those established in the initial experiment. K in the KNN model has a value of 7. The MaxEnt model's maximum step length is set to 10 (Max step = 10), and the probability distribution adopts empirical edge distribution probability, which is optimized by a geographic information system (GIS) algorithm. In the CS-BP model, the population size is set to twenty, and the probability of discovery is set to 0.25. Other models' parameters retain the default values of MATLAB tools.

It is important to avoid individual classifiers from achieving the local optimum, i.e., [normal, fault] = [0, 1], and thus, producing "false positives" for the fault categories. This experiment also provides the expected diagnosis rate for the normal condition. Table 8 shows the expected diagnosis rates of different classifiers for both normal and problem states.

**Table 8.** Predicted diagnostic rates of normal states and fault state corresponding to each fault class for different classifiers (N represents normal states, F corresponds to fault states).

| Category | MSSA-PNN | | LDA | | QDA | | KNN | | SVM | | MaxEnt | | CS-BP | |
|---|---|---|---|---|---|---|---|---|---|---|---|---|---|---|
| | N | F | N | F | N | F | N | F | N | F | N | F | N | F |
| 1 | 1.00 | 1.00 | 1.00 | 0.99 | 1.00 | 0.99 | 0.51 | 0.83 | 1.00 | 0.99 | 1.00 | 0.89 | 1.00 | 0.99 |
| 2 | 0.96 | 0.99 | 1.00 | 0.95 | 1.00 | 0.98 | 0.17 | 0.81 | 1.00 | 0.98 | 0.74 | 0.29 | 0.99 | 0.97 |
| 3 | 0.96 | 0.98 | 0.55 | 0.54 | 0.78 | 0.50 | 0.39 | 0.82 | 0.00 | 1.00 | 0.84 | 0.11 | 0.46 | 0.84 |
| 4 | 1.00 | 1.00 | 1.00 | 1.00 | 1.00 | 1.00 | 0.30 | 0.86 | 1.00 | 1.00 | 0.83 | 0.59 | 1.00 | 1.00 |
| 5 | 1.00 | 1.00 | 1.00 | 0.99 | 1.00 | 1.00 | 0.23 | 0.81 | 1.00 | 1.00 | 0.79 | 0.18 | 1.00 | 1.00 |
| 6 | 1.00 | 0.99 | 1.00 | 0.96 | 1.00 | 0.99 | 0.99 | 0.97 | 1.00 | 1.00 | 1.00 | 0.24 | 0.99 | 0.99 |
| 7 | 1.00 | 1.00 | 1.00 | 1.00 | 1.00 | 1.00 | 0.44 | 0.77 | 1.00 | 1.00 | 0.89 | 0.19 | 1.00 | 1.00 |
| 8 | 0.75 | 0.83 | 0.69 | 0.50 | 0.97 | 0.90 | 0.18 | 0.81 | 0.00 | 1.00 | 0.83 | 0.22 | 0.95 | 0.90 |
| 9 | 0.72 | 0.87 | 0.53 | 0.61 | 0.54 | 0.61 | 0.32 | 0.75 | 0.00 | 1.00 | 0.84 | 0.58 | 0.25 | 0.88 |
| 10 | 0.73 | 0.82 | 0.63 | 0.47 | 0.80 | 0.57 | 0.35 | 0.81 | 0.00 | 1.00 | 0.93 | 0.53 | 0.29 | 0.81 |
| 11 | 0.66 | 0.84 | 0.50 | 0.52 | 0.71 | 0.55 | 0.30 | 0.74 | 0.00 | 1.00 | 0.84 | 0.18 | 0.46 | 0.76 |
| 12 | 0.76 | 0.94 | 0.73 | 0.50 | 0.98 | 0.95 | 0.63 | 0.80 | 0.01 | 0.67 | 0.87 | 0.10 | 0.96 | 0.94 |
| 13 | 0.97 | 0.98 | 0.80 | 0.57 | 1.00 | 0.97 | 0.87 | 0.88 | 0.23 | 0.60 | 0.91 | 0.18 | 0.98 | 0.97 |
| 14 | 1.00 | 1.00 | 0.51 | 0.50 | 1.00 | 1.00 | 0.93 | 0.93 | 0.00 | 1.00 | 0.97 | 0.15 | 1.00 | 1.00 |
| 15 | 0.64 | 0.82 | 0.58 | 0.61 | 0.68 | 0.52 | 0.30 | 0.83 | 0.00 | 1.00 | 0.87 | 0.54 | 0.13 | 0.95 |
| 16 | 0.64 | 0.86 | 0.51 | 0.52 | 0.69 | 0.49 | 0.36 | 0.84 | 0.00 | 1.00 | 0.95 | 0.58 | 0.22 | 0.85 |
| 17 | 0.79 | 0.87 | 0.48 | 0.56 | 0.79 | 0.61 | 0.30 | 0.80 | 0.00 | 1.00 | 0.89 | 0.26 | 0.56 | 0.82 |
| 18 | 0.77 | 0.88 | 1.00 | 0.81 | 1.00 | 0.87 | 0.87 | 0.86 | 1.00 | 0.88 | 0.96 | 0.62 | 0.95 | 0.88 |
| 19 | 0.63 | 0.86 | 0.53 | 0.52 | 0.65 | 0.53 | 0.23 | 0.81 | 0.00 | 1.00 | 0.74 | 0.24 | 0.22 | 0.89 |
| 20 | 0.51 | 0.80 | 0.52 | 0.57 | 0.79 | 0.59 | 0.40 | 0.82 | 0.00 | 1.00 | 0.81 | 0.34 | 0.55 | 0.83 |
| 21 | 0.74 | 0.84 | 0.67 | 0.55 | 0.90 | 0.72 | 0.34 | 0.79 | 0.00 | 1.00 | 0.74 | 0.28 | 0.71 | 0.87 |
| Mean | 0.82 | 0.91 | 0.72 | 0.68 | 0.87 | 0.78 | 0.45 | 0.83 | 0.35 | 0.96 | 0.87 | 0.35 | 0.70 | 0.91 |

Table 8 shows that the fault diagnosis rate for MSSA-PNN is 91%, and 68% for LDA, 78% for QDA, 83% for KNN, 96% for SVM, 35% for MaxEnt, and 91% for CS-BP. If we only look at the rate of diagnosing each fault state, the diagnostic performance of the MSSA-PNN model is almost the same as that of the CS-BP model and is even worse than that of the SVM model. Table 8 also shows that SVM has "false positives" in 14 fault categories, such as categories 3, 8, 9, 10, and 11. Category 9, 10, 15, 16, and 19 faults have "local optimization" in CS-BP. Thus, judging the diagnostic performance of different classifiers only by the rate at which each fault category is found is unfair and insufficient.

In order to completely analyze the defect diagnostic performance of all classifiers and demonstrate the efficacy of the proposed model, the study chose the accuracy rate and F1-score as evaluation indices. The confusion matrix is a crucial metric for assessing the

performance of classification models. Table 9 shows that it has four values: true positive (TP), true negative (TN), false positive (FP), and false-negative (FN). TP denotes the number of correctly predicted positive samples; TN denotes the number of correctly predicted negative samples; FP denotes the number of predicted positive samples that are actually negative; FN denotes the number of predicted negative samples that are actually positive.

Precision refers to the ratio between the number of samples that are correctly predicted as positive labels and the total number of samples that are predicted as positive labels, as shown in Equation (21):

$$\text{precision} = \frac{TP}{TP + FP} \tag{21}$$

The recall is the ratio of the number of samples that were correctly predicted to have positive labels to the number of samples that were labeled positive, as shown in Equation (22):

$$\text{recall} = \frac{TP}{TP + FN} \tag{22}$$

The two key evaluation indicators required in this paper can also be obtained from Table 9, and the calculation formula is as follows:

$$\text{accuracy} = \frac{TP + TN}{TN + TP + FN + FP} \tag{23}$$

$$\text{F1-score} = \left(1 + \beta^2\right) \frac{\text{precision} \times \text{recall}}{\left(\beta^2 \cdot \text{precision}\right) + \text{recall}} \tag{24}$$

In this paper, $\beta = 1$, indicating that precision and recall are considered with the same weight.

**Table 9.** Confusion matrix for evaluating machine learning.

| Actual Class | Predicted Class | |
|---|---|---|
| | **Positive** | **Negative** |
| Positive | True positive (TP) | False Negative (FN) |
| Negative | False positive (FP) | True Negative (TN) |

The accuracy of MSSA-PNN and other classifiers is displayed in Table 10. MSSA-PNN's diagnostic accuracy is 88%, which is greater than LDA's 69%, QDA's 81%, KNN's 70%, SVM's 75%, MaxEnt's 52%, and CS-BP's 84%. In addition, MSSA-PNN has a considerable advantage over other classifiers in eight fault categories: category 3, category 9, category 10, category 11, category 15, category 16, and category 19. Although it falls short in a few defect categories, the disparity is negligible. Therefore, MSSA-PNN provides more accurate fault diagnosis performance.

The F1-score of MSSA-PNN and other classifiers are displayed in Table 11. From comparing Tables 8 and 11, it is evident that certain classifiers may have a greater rate of fault diagnosis than MSSA-PNN. However, the F1-score is lower than the MSSA-PNN score. For instance, in the SVM's categories 8, 9, 14, etc., the fault diagnosis rates are more significant than those of MSSA-PNN, but their F1-score is significantly lower than that of MSSA-PNN. It shows that SVM's category 8 and other fault categories display over-fitting. Consequently, the F1-score and the accuracy rate can be used together to give a complete picture of the classifier's ability to find faults. Meanwhile, the average F1-score of MSSA-PNN is 91%, which is higher than that of other classification models, while LDA is at 74%, QDA is at 83%, KNN is at 79%, SVM is at 84%, MaxEnt is at 46%, and CS-BP is at 88%.

In conclusion, based on the results, the fault diagnosis rate, accuracy rate, and F1-score of MSSA-PNN are the same, indicating that there is nearly no overfitting issue with this model, the results are trustworthy, and the diagnosis model is persuasive. In general,

the model suggested by this research does a better job of diagnosing problems than some commonly used classifiers.

**Table 10.** The accuracy of MSSA-PNN and other classification models.

| Category | MSSA-PNN | LDA | QDA | KNN | SVM | MaxEnt | CS-BP |
|---|---|---|---|---|---|---|---|
| 1 | 1.00 | 0.99 | 1.00 | 0.72 | 0.99 | 0.93 | 0.99 |
| 2 | 0.98 | 0.97 | 0.99 | 0.59 | 0.99 | 0.44 | 0.98 |
| 3 | 0.98 | 0.54 | 0.59 | 0.67 | 0.66 | 0.36 | 0.71 |
| 4 | 1.00 | 1.00 | 1.00 | 0.67 | 1.00 | 0.68 | 1.00 |
| 5 | 1.00 | 0.99 | 1.00 | 0.62 | 1.00 | 0.39 | 1.00 |
| 6 | 0.99 | 0.98 | 1.00 | 0.98 | 1.00 | 0.49 | 0.99 |
| 7 | 1.00 | 1.00 | 1.00 | 0.66 | 1.00 | 0.42 | 1.00 |
| 8 | 0.80 | 0.56 | 0.92 | 0.59 | 0.66 | 0.41 | 0.92 |
| 9 | 0.82 | 0.58 | 0.59 | 0.62 | 0.65 | 0.67 | 0.67 |
| 10 | 0.79 | 0.52 | 0.64 | 0.65 | 0.65 | 0.66 | 0.63 |
| 11 | 0.78 | 0.51 | 0.60 | 0.60 | 0.67 | 0.41 | 0.66 |
| 12 | 0.88 | 0.58 | 0.96 | 0.74 | 0.44 | 0.37 | 0.95 |
| 13 | 0.98 | 0.65 | 0.98 | 0.88 | 0.47 | 0.44 | 0.97 |
| 14 | 1.00 | 0.51 | 1.00 | 0.93 | 0.66 | 0.44 | 1.00 |
| 15 | 0.76 | 0.60 | 0.57 | 0.65 | 0.66 | 0.66 | 0.64 |
| 16 | 0.78 | 0.52 | 0.56 | 0.68 | 0.65 | 0.71 | 0.63 |
| 17 | 0.84 | 0.53 | 0.67 | 0.63 | 0.65 | 0.49 | 0.74 |
| 18 | 0.84 | 0.88 | 0.91 | 0.86 | 0.92 | 0.74 | 0.90 |
| 19 | 0.79 | 0.53 | 0.57 | 0.62 | 0.63 | 0.40 | 0.65 |
| 20 | 0.71 | 0.56 | 0.66 | 0.67 | 0.66 | 0.49 | 0.74 |
| 21 | 0.81 | 0.59 | 0.78 | 0.64 | 0.65 | 0.43 | 0.82 |
| Mean | 0.88 | 0.69 | 0.81 | 0.70 | 0.75 | 0.52 | 0.84 |

**Table 11.** The F1-score of MSSA-PNN and other classification models.

| Category | MSSA-PNN | LDA | QDA | KNN | SVM | MaxEnt | CS-BP |
|---|---|---|---|---|---|---|---|
| 1 | 1.00 | 0.99 | 1.00 | 0.80 | 1.00 | 0.94 | 1.00 |
| 2 | 0.98 | 0.97 | 0.99 | 0.72 | 0.99 | 0.41 | 0.98 |
| 3 | 0.98 | 0.61 | 0.63 | 0.77 | 0.79 | 0.18 | 0.79 |
| 4 | 1.00 | 1.00 | 1.00 | 0.78 | 1.00 | 0.70 | 1.00 |
| 5 | 1.00 | 1.00 | 1.00 | 0.74 | 1.00 | 0.28 | 1.00 |
| 6 | 1.00 | 0.98 | 1.00 | 0.98 | 1.00 | 0.39 | 0.99 |
| 7 | 1.00 | 1.00 | 1.00 | 0.75 | 1.00 | 0.30 | 1.00 |
| 8 | 0.85 | 0.61 | 0.94 | 0.72 | 0.80 | 0.33 | 0.94 |
| 9 | 0.87 | 0.66 | 0.66 | 0.73 | 0.79 | 0.69 | 0.78 |
| 10 | 0.84 | 0.57 | 0.68 | 0.75 | 0.79 | 0.68 | 0.74 |
| 11 | 0.83 | 0.59 | 0.65 | 0.71 | 0.80 | 0.28 | 0.75 |
| 12 | 0.91 | 0.61 | 0.97 | 0.80 | 0.61 | 0.16 | 0.96 |
| 13 | 0.98 | 0.68 | 0.98 | 0.90 | 0.60 | 0.30 | 0.98 |
| 14 | 1.00 | 0.57 | 1.00 | 0.95 | 0.79 | 0.26 | 1.00 |
| 15 | 0.82 | 0.67 | 0.61 | 0.76 | 0.80 | 0.67 | 0.77 |
| 16 | 0.84 | 0.59 | 0.59 | 0.77 | 0.79 | 0.73 | 0.75 |
| 17 | 0.88 | 0.61 | 0.71 | 0.74 | 0.79 | 0.40 | 0.80 |
| 18 | 0.87 | 0.89 | 0.93 | 0.89 | 0.93 | 0.75 | 0.93 |
| 19 | 0.85 | 0.59 | 0.62 | 0.74 | 0.78 | 0.35 | 0.77 |
| 20 | 0.79 | 0.63 | 0.69 | 0.76 | 0.80 | 0.48 | 0.81 |
| 21 | 0.85 | 0.64 | 0.81 | 0.74 | 0.79 | 0.39 | 0.86 |
| Mean | 0.91 | 0.74 | 0.83 | 0.79 | 0.84 | 0.46 | 0.88 |

## 4. Conclusions

This paper develops a basic fault diagnosis model based on feature selection and PNN. The main innovation of this paper is to use PSO-SVM to pre-screen the SVM-RFE feature selection and MSSA to optimize the PNN. PSO-SVM-RFE performs feature selection by two-

level screening, which can remove redundant features, simplify the samples, and indirectly improve the classification performance. MSSA uses a unique optimization mechanism to update the parameters, giving the algorithm better global search capabilities.

In this paper, the fault diagnosis performance of the PSO-SVM-RFE-MSSA-PNN model is experimentally validated using experimental data provided during the TE chemistry process. The analysis of the experimental results shows that the PSO-SVM-RFE feature selection method can improve the classification accuracy, and MSSA can enhance the local convergence of PNN, making the combined model have better fault diagnosis performance. Therefore, PSO-SVM-RFE-MSSA-PNN is suitable for fault prediction and diagnostic classification of the Tennessee Eastman process.

Although the model proposed in this paper has achieved good results to some extent, there are still some limitations that need further improvement in future work:

1. This paper uses a two-stage feature selection algorithm to delete redundant features. Experiments verify the practicability and superiority of the algorithm, but the influence of operation time is ignored in the experiments. In future work, further simplification of the structure of the feature selection algorithm, such as adopting NSGA-II, should be considered to achieve simplification in the data preprocessing stage.
2. The quality of the original data of the TE process directly affects the diagnostic performance of the fault diagnosis model. By observing the sample data, it can be seen that each characteristic variable of the TE process fluctuates in a normal state, which can easily cause misjudgment, which may be the reason for the low diagnosis rate of some fault types. Therefore, it is necessary to scale out abnormal data before feature selection in practical applications.
3. The feature selection algorithm in this paper only filters redundant features at the data level. Next, we can combine the characteristics of the TE chemical process itself, explore the chemical connection between the characteristic variables, and ignore unnecessary variables from the chemical direction, which will be a new cross-optimization direction.
4. In a natural chemical process, faults can be divided into process and sensor faults. Process faults are characterized by multivariate coordination, while sensor faults are variable independent, and the fault variable is unique. The occurrence of a process fault means that the system's operating state deviates from its normal value. In contrast, sensor faults interfere with the system's stability and affect the operator's judgment, which may lead to failures. Examples are drift, jitter, and stepping of data. This paper makes no distinction between the two, but they are uniformly classified as faults. Therefore, we should distinguish and differentiate between process and sensor faults in chemical processes in subsequent research.

**Author Contributions:** Conceptualization, H.X.; methodology, H.X.; software, H.X.; validation, Z.M. and T.R.; formal analysis, H.X.; writing—original draft preparation, H.X.; writing—review and editing, X.Y.; visualization, Z.M. and T.R.; funding acquisition, X.Y. All authors have read and agreed to the published version of the manuscript.

**Funding:** This work was supported in part by the National Natural Science Foundation of China (51765042, 61963026).

**Institutional Review Board Statement:** Not applicable.

**Informed Consent Statement:** Not applicable.

**Data Availability Statement:** Not applicable.

**Conflicts of Interest:** The authors declare no conflict of interest.

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
