# Peer review of "A Fault Diagnosis Model for Tennessee Eastman Processes Based on Feature Selection and Probabilistic Neural Network"

_applsci, doi:10.3390/app12178868_

Round 1
Reviewer 1 Report (Previous Reviewer 3)
The submission contains the original concepts and work of the Authors, which may be useful for other researchers dealing with fault detection and isolation problems. However, in my opinion, the quality of presentation should be substantially improved before acceptance of the submission for publication.
The main comments:
-- Description of the tools and algorithms used by the Authors to build the classifiers occupies almost half of the text volume, which (in my opinion) is too much.
-- Fig. 2 (page 6) - the split into the training and the test subsets should be mrked (in the left part of the figure) after the feature selection stage. Now it may suggest that not the same features are chosen for those two subsets.
-- Section 2.6 (page 10) describes the design process of the PSO-SVM-RFE-MSSA-PNN algorithm before (!) the problem studied in the submission was formally stated. Also the term 'TE' should be explained somewhere before using only the abbreviation.
-- The experimental setup (for simulations) should be described in a more especific and detailed way.
-- In Tables 3-5 are the numbers of the featured. Do the numbers correspond to the numbers of variables form Table 2? If so, what's the variable No. 53? If not, what's the physical (or mathematical) meaning of the features?
-- Why only 6 features were selected as inputs to the classifiers? i am sorry, but I could not find this information in the text.
-- The categories of faults in Tables 3-5 are the same as mentioned in Table 1, as I understand. So, how did the Authors collect data for faults in categories 16-20, marked as "Unknown" in Table 1? And how they distinguished those categories?
-- The Authors compare their work to the results given by other classifiers. But they do not mention what were the inputs to those other classifiers. If the same to their own classifiers, one should ask about the quality of such an approach. Maybe the othe methods can give better results for different selection of the input set?
Author Response
Please see the attachment

Reviewer 2 Report (New Reviewer)
There are several queries need to be address by authors :
1. Title of manuscript should be changed ? what is the meaning of TE ? Further word novel should be discarded as there is no relevance of this word based on methodology proposed.
2. Manuscript seems to be very lengthy, and it is suggested to include necessary details in brief and with relevant discussions only.Specifically PSO and salp swarm algorithm.
3. Justification needed to select kernel function and penalty parameter values chosen in SVM.
4. It is recommended to show the utility of feature selected either qualitatively or quantitatively. Refer :
a. https://link.springer.com/article/10.1007/s00500-015-1608-6.
5. What is the rationale to choose recursive feature elimination and PSO when other feature selection/ranking method like Fisher score,ReliefF,Information Gain,Mutual information are utilized by various authors.Kindly add the discussion in revised manuscript.
6. The heading of section 4 should be modified instead of CONCLUSIONS and DISCUSSION it should be named as CONCLUSION only. Further unnecessary details should be deleted.
7. Resolution of Fig.3,5,6 should be improved for better readability.
8. It is suggested to modify manuscript after careful reading since facts and findings are repeated in various sections.
3.
Author Response
Please see the attachment.

Reviewer 3 Report (New Reviewer)
The authors must compare the MSE and RMSE for several techniques compared in this article in order to be able to show the precision, the reliability and the speed of their method.
What type of data and how much data was used in the PNN structure?
In this study the authors did not study their proposed method with LSTM+Genetic algorithm NSGA II. Why? this study will be useful.
Figures 7 and 8 must be analyzed and explained in which case and under which condition these faults occur? these faults are transient faults or permanent faults?
The structure of the article must be modified certain parts require a deep revision among others: tables 3, 4 and 5.
The discussion part and the conclusion must be separated.
The authors can use the articles below to improve their article.
- A Fault Diagnosis Design Based on Deep Learning Approach for Electric Vehicle Applications, MDPI Energies 2021
- Machine learning (ML) algorithms and artificial neural network for optimizing in vitro germination and growth indices of industrial hemp (Cannabis sativa L.), ELSEVIER, Industrial Crops and Products, July 2022.
- Fault Diagnosis of Smart Grids Based on Deep Learning Approach, WAC2021
- Accuracy of predictions made by machine learned models for biocrude yields obtained from hydrothermal liquefaction of organic wastes, Chemical Engineering Journal, August 2022
Author Response
Please see the attachment.

Reviewer 4 Report (New Reviewer)
The article presents an interesting and innovative (although it is mostly a combination of known solutions) algorithm for the isolation of faults designed for complex processes for which extensive archival data sets are available. The operation of the algorithm has been thoroughly examined on the example of data from the Tennessee Eastman process. The results obtained were thoroughly, thoroughly and correctly described and evaluated.
Mainly, one can have a reservation about the practical application of the proposed method, because it requires the possession of teaching data for states with faults. In the introduction, the authors criticize other approaches (including “classical pattern recognition" methods) as not adapted to this type of complex processes. Unfortunately, at the same time, they were silent about the problem of the need to have teaching data from states with faults. Unfortunately, in most cases, this is a requirement that makes it practically impossible to use such an approach, which also belongs to the group of "pattern recognition" methods. As a result, the presented considerations are somewhat academic in nature.
Also the problem of process dynamics (visible in data timeseries) is not sufficiently addressed.
The above remarks do not detract from the complexity of the research carried out and from the precise and prudent formulation of the final conclusions.
The article has a well-chosen structure and is quite carefully written.
Many detailed comments can be found in attached pdf file.

Author Response
Please see the attachment.

Reviewer 5 Report (New Reviewer)
The author proposes a Tennessee Eastman process fault diagnosis model based on feature selection and probabilistic neural network. The authors' results show that PSO-SVM-RFE feature selection of data samples simplifies and eliminates redundant features more effectively than other feature selection techniques. The results of the paper look both reasonable and valuable. However, there are several important aspects that require authors' comments or possibly improvements:
l Reviewer noticed that the paper is similar to the author's prior work:
Tao, L.; Yang, X.; Zhou, Y.; Yang, L. A Novel Transformers Fault Diagnosis Method Based on Probabilistic Neural Network and Bio-Inspired Optimizer. Sensors 2021, 21, 3623. https://doi.org/10.3390/s21113623
Considering this issue, the reviewer kindly asks that author clarify the novelty and enhancements between them.
l The bibliography should be significantly expanded. the reviewer recommends adding a literature review with new references. The latest references are from 2020. It is recommended to add references in MDPI.
l The reviewers recommend that the authors add parameters to the proposed method in the Results Discussion and describe how these parameters were chosen.
l It is recommended that the authors' approach increases the limitations of the study in the conclusion.
Author Response
Please see the attachment.

Reviewer 6 Report (New Reviewer)
In this work is presented a novel chemical process defect diagnosis model based on Tennessee Eastman data and prior research expertise. The work has been improved significantly and it can be considered for its publication in its present form.
Author Response
首先,我们要感谢审稿人花时间审稿,并感谢他们给予了积极的评价。
Round 2
Reviewer 1 Report (Previous Reviewer 3)
The second version of the submission is better than the previous one. The quality of presentation has been improved, so the submission may be considered for publication. Some of the issues mentioned in the previous review has been explained by the Authors in the cover letter and improved in the re-submitted manuscript.
Anyway, I still think that the paper is too long. For example - what's the use of presenting figures 6 nad 7, if the points on the 'Reaction process time' axis say nothing to the reader, especially when the graphs are poorly commented in the text?
There are also some grammar or spelling mistakes in the text, so some help from a native speaker would be beneficial.
Author Response
Please see the attachment.

Reviewer 2 Report (New Reviewer)
Authors address all reviewer comments and modified manuscript accordingly.
Author Response
We are grateful to the reviewers for their approval of the manuscript and their comments in the first round of review, which have played a positive role in making better revisions to the manuscript. In addition, we note another request from the reviewers that the English language and style are acceptable/minor spell check is required because English is not our native language, so there may be some inappropriate expressions in the manuscript, but we have made a full English correction in this revision. We hope to get the approval of the reviewers here. If there are any areas that the reviewers feel have not been adequately addressed, we hope they will point out the problems, and we thank them again.
Reviewer 3 Report (New Reviewer)
The article suffers from a real organizational problem.
Several parts are long without any common thread between the parts.
The authors have not taken into account all the details requested by the reviewer.
The article ends with a discussion without a conclusion!!!!
Author Response
We responded point by point with a combination of the reviewer's letter to the editor and the second round of review comments, and we re-responded to the first round of review comments. We have edited both in a PDF. Please see the attachment.

This manuscript is a resubmission of an earlier submission. The following is a list of the peer review reports and author responses from that submission.
Round 1
Reviewer 1 Report
The author solves an interesting problem of fault diagnosis by using data from industrial chemical processing. The used data include 960 samples, among which 160 samples are used for training. The author combines several typical algorithms together, and try to improve the classification performance. In the paper, the features are selected by PSO-SVM-RFE method, then the fault are diagnosed by PNN which is optimized by MSSA. Lot of simulation experiments has done to show the efficiency of the proposed method.
However, the contribution is not clear enough, the novelty of the paper is weak, the organization of the paper should be improved, the written skill should be polished to be a published Journal paper.
Reviewer 2 Report
This paper proposes a fault diagnosis method for the industrial chemical process under the problem of the high correlation between process characteristic variables. The proposed method employs biological source optimizers to derive the best performance of the PNN classifier. The proposed method is validated based on the experimental data from Tennessee Eastman (TE) process. Overall, this paper is not qualified for publication. The reviewer’s main concerns are lack of novelty of the proposed method. The detailed commends are shown as follows.
1. Overall, it is highly recommended to polish English writing with the help of English experts or professional editors. In the current manuscript, there have been found many grammar errors, awkward expressions, and misused punctuation marks.
2. In the Introduction, this paper addresses the problem of correlated features in terms of removing redundant features. However, the proposed method only combines several classifiers (i.e., SVM and PNN) and optimizers (i.e., PSO and MSSA). How could the proposed method solve the addressed problem?
3. The reviewer thinks that the title of this paper cannot include the proposed method. The authors described the proposed method based on a PSO-SVM-REF-MSSA-PNN. Therefore, it is recommended to make the title more specific to describe the proposed method.
4. It is recommended to reorganize Section 2 because this section describes the proposed method. However, too many subsections reduce the readability of this section. Please address this issue.
5. The reviewer wonders why the weights in Eq. (10) are set as 0.2 and 0.8. Without the proper reason, this setting cannot guarantee the generalizability of the proposed method. Please address this issue.
6. It seems that the PSO-SVM method only optimizes the hyperparameters of the SVM. Then, how could this method filter the redundant features?
7. The authors have not suggested the future work in Section 4. Please address this issue.
8. Several minor comments are suggested below.
A. What is the difference between support future machine and support vector machine?
B. Eq. (5) is described without the definition of each term.
C. The description of Figure 6 is followed after the descriptions of Figures 7 and 8.
D. Please double-check Category in Table 10.
Reviewer 3 Report
1. The review could present the state-of-the-art in a more systematic way, presenting the most commonly used approaches, their characteristics, advantages and drawbacks, etc.
2. I have scored the 'Quality of presentation' as 'average', for example because:
-- feature sets for different methods, presented in Tables 3-5, are not possible to be compared, so what's the reason of showing them?
-- parformances for different faults (figures 7-8) occupy two pages for no apparent reason;
-- the Authors describe the results presented in a form of tables or figures, but they do not try to generate any explanation of such results.
There are also some editorial erros, e.g.:
-- page #11, in description of 'Step 9' there is return to the same step;
-- formulas (28) and (29) seem to be the same;
-- it should be 'F1-score' (without spaces) in formula (30).
There are also many small editorial mistakes, which should be improved.